# Evaluating the impact of COVID-19 on routine childhood immunizations coverage in Zambia

**Kelvin Mwangilwa**[1]*, **Charles Chileshe**[1], **John Simwanza**[1], **Musole Chipoya**[1], **Davie Simwaba**[1], **Nathan Kapata**[1], **Mazyanga Lucy Mazaba**[1], **Nyuma Mbewe**[1], **Kapina Muzala**[1], **Nyambe Sinyange**[1], **Isaac Fwemba**[2], **Roma Chilengi**[1]

**1** Zambia National Public Health Institute, Kabulonga, Lusaka, **2** Department of Epidemiology and Biostatistics, University of Zambia, Lusaka, Zambia

* kelvinpetersj@gmail.com

**Data Availability Statement:** The data is available online deposited in figshare repository and the link to the file is https://doi.org/10.6084/m9.figshare.26190104.v1.

## Abstract

There are growing concerns about the comeback of vaccine-preventable diseases. Epidemics exert shocks which affect other health performance indicators such as routine immunizations. Early model forecasts indicate decreased use of immunization services, which puts children at greater risk. Concerns about an increase in morbidity and mortality for illnesses other than COVID-19, particularly in children missing routine vaccinations, are of public health interest. In this study, we evaluate COVID-19 effects on the uptake of routine immunization in Zambia. This was an interrupted time series study. National data on routine immunization coverage between January 2017 and December 2022 were analyzed. Interrupted time series analysis was performed to quantify changes in immunization utilization. To determine if changes in the underlying patterns of utilization of immunization service were correlated with the commencement of COVID-19, seasonally adjusted segmented Poisson regression model was utilised. Utilization of health services was similar with historical levels prior to the first case of COVID-19. There was a significant drop in immunization coverage for measles dose two (RR, 0.59; 95% CI: 0.43–0.80). A decreased slope was observed in immunization coverage of Rotavirus dose one (RR, 0.97; 95% CI: 0.96–0.98) and Rotavirus dose two (RR, 0.97; 95% CI: 0.96–0.98). A growing slope was observed for Oral Poliovirus two (RR, 1.007; 95% CI: 1.004–1.011) and Oral Poliovirus three (RR, 1.007; 95% CI: 1.002–1011). We also observed a growing slope in BCG Bacille Calmette-Guerin (BCG) (RR, 1.001; 95% CI: 1.000–1011) and Pentavalent one (RR, 1.00; 95% CI: 1.001–1008) and three (RR, 1.004; 95% CI: 1.001–1008). The COVID-19 pandemic has had a number of unintended consequences that have affected the use of immunization services. Ensuring continuity in the provision of health services, especially childhood immunization, during pandemics or epidemics is crucial. Therefore, Investing in robust healthcare infrastructure to withstand surges, training and retaining a skilled workforce capable of handling emergencies and routine services simultaneously is very cardinal to avoid vaccine-preventable diseases, causing long-term health effects especially child mortality.

**Funding:** The authors received no specific funding for this work.

**Competing interests:** The authors have declared that no competing interests exist.

## Introduction

The World Health Organization (WHO) declared the novel COVID-19 outbreak a global pandemic on March 11, 2020. Globally, 6.8 million fatalities and 758 million cases have been reported as of February 28, 2023 [1]. Healthcare services continue to be impacted by the current COVID-19 worldwide pandemic beyond those that are specifically dedicated to the control of SARS-CoV-2 transmission and sickness [2]. Most regions and nations have been affected by widespread global disruption in normal childhood immunization, particularly in the early pandemic periods [3,4]. However, according to WHO and United Nations International Children's Emergency Fund (UNICEF) estimates, immunization coverage in Africa had been gradually increasing over the years before the pandemic. For instance, the coverage for the third dose of the diphtheria-tetanus-pertussis (DTP3) vaccine had reached about 74% by 2019 [5,6]. Measles-containing vaccine (MCV1) coverage also showed improvement, reaching around 69% by 2019 [7,8]. In spite of the improvements, there are growing concerns about the comeback of vaccine-preventable diseases, particularly measles, because the downstream impact may be even worse in resource-limited settings and economically impoverished populations [9].

By 3 February 2023, Zambia recorded 339, 092 cases and 4, 040 deaths since the first COVID-19 case in Zambia was reported on March 18, 2020 [10]. A wide range of interventions were put in place by the government in an effort to stop the pandemic's spread [11,12]. The COVID-19 pandemic's direct effects on population health and also indirect effects brought on by control measures, alterations in the behavior of patients and healthcare professionals, and reorganization of the health system are likely to result in changes in how people utilize immunizations [13].

Economic and social effects such as decreased manufacturing, easier access to jobs, and reduced costs for essential commodities are also anticipated as a result of the public health efforts [14]. Indeed access to, and use of crucial health services are therefore likely to be impacted [15]. For example Early model forecasts indicated decreased use of health services [13,14]. Studies conducted during earlier epidemics in sub-Saharan Africa also revealed a decline in access and utilization of key medical services both during and after outbreaks [16]. Different population groups are probably affected differently, with women and children being more at risk [17]. Concerns about an increase in morbidity and mortality for illnesses other than COVID-19, particularly for child health services, are being raised as a result of these pauses in the use of health services [18]. The impact of the pandemic on the delivery of vaccines and the monitoring of a wide range of essential services over a longer period of observation after the pandemic as announced by the WHO, has not been thoroughly evaluated in Zambia. There is evidence that in Africa there was a shortage in rotavirus during the COVID-19 pandemic [19–21]. Nonetheless other recent studies within the region have reported variable impacts on different health services [22].

This study sought to determine the effects of COVID-19 on routine childhood immunizations coverage in Zambia.

## Methods

### Study design

The study utilized longitudinal design using the national routine immunization data. Interrupted time series analysis was performed to quantify changes in immunization utilization with adjusted segmented Poisson regression model. We compared Zambia monthly immunization rates before and after the COVID-19 pandemic. The period running from January 2018 to February 2020 just before the first reported case was defined as pre-COVID-19; and March

2020 to December 2021 as post-COVD-19. These data were deemed appropriate as it was collected at monthly intervals and is consistent with interrupted time series designs[22,23] We used it to determine an overarching trend, and considered COVID-19 as an intervention that took place from March 2020. We thus considered the period after as post-intervention to establish the trends. We then made comparisons on the trends using the preexisting trends as counterfactual [24].

We analyzed coverage rates of four vaccine antigens in our routine child immunization program (Measles 1 & 2; Pentavalent 1, 2 & 3, Rota 1 & 2 and Oral Poliovirus Vaccine 1, 2 &3) as well as fully immunized variables.

## Data sources

We accessed the national data which is collected from all healthcare facilities using the District Health Information Software 2 (DHIS2) for the Health Management Information System (HMIS) guided by the National essential indicator set (EIS) of the Ministry of Health. Health records and information officers located at various health facilities enter monthly aggregated facility level data into the system using a range of instruments, including desktop computers, laptops, tablets, and cellphones. Data is routed to a central administrative unit where it is pooled and entered into the national database. Further, Immunization data entered in the DHIS2 is collected using facility-based child health immunization register, aggregated by the health facility staff on a monthly basis using a standard paper-based service delivery aggregation form 2 (HIA2). The District Health Office (DHO) then validates and enters the data into a national online based DHIS2 platform. The acceptable report completeness rate for vaccination is an average of 80% every month. The data shows every month it was over the threshold for completeness and the mean value was 94.0% of complete report rate during the period of observation [25]. The completeness improved over time including the pandemic period (see Table 1). Further, we used static population estimates provided by the Zambia Statistics Agency (ZamStats) from the 2010 projected census population estimates to account for the denominators and estimate coverage, which changed from one year to another year due to population growth with each month having the same denominator by dividing the projected population for each year by twelve months [26]. The percentage of age-eligible children who were immunized by 30 or 31 days after the due date (28 or 29 days after the February due dates) was used to define vaccination coverage, as has been reported elsewhere [27,28]. The population is based on the yearly projected population divided by 12 months, as the

**Table 1. Data completeness from 2017 to 2022 by month.**

| Month | 2017 | 2018 | 2019 | 2020 | 2021 | Feb-22 |
|---|---|---|---|---|---|---|
| January | 86.9 | 87.9 | 86.9 | 96.3 | 91.8 | 98.9 |
| February | 84.4 | 88.4 | 84.4 | 96.2 | 96.6 | 99.3 |
| March | 84.1 | 88.1 | 84.1 | 96.2 | 97.2 | 99.2 |
| April | 84.3 | 94.3 | 94.3 | 96.4 | 97.5 | 98.2 |
| May | 84.2 | 94.2 | 94.2 | 96.1 | 97.6 | 99.2 |
| June | 90.4 | 94.4 | 94.4 | 96.1 | 97.3 | 98.9 |
| July | 89.2 | 94.1 | 94.1 | 96.3 | 97.4 | 99.2 |
| August | 84.5 | 94.5 | 94.5 | 96.4 | 97.9 | 99.1 |
| September | 89.6 | 94.6 | 94.6 | 96.1 | 98.1 | 99.3 |
| October | 90.0 | 95.0 | 95.0 | 94.2 | 96.1 | 99.2 |
| November | 90.0 | 95.0 | 95.0 | 95.5 | 95.8 | 99.0 |
| December | 89.8 | 95.0 | 95.0 | 95.6 | 97.3 | 99.5 |

population are not assumed to change from one month to the other but on yearly basis. Our data is also routinely administered and recorded, therefore relied on what was reported in the system on each vaccine.

In order to calculate the mean monthly rate we captured trends in eligible populations and corresponding coverage, we included monthly vaccination patterns in 2017 through to 2022 [3]. For each vaccination dose from January 2017 through December 2022, we evaluated the monthly coverage. Comparisons between coverage in each month of 2022 and the corresponding month of 2017 were made. Each month, was examined separately. All children who were age-eligible for vaccination were captured at heath facilities. In order to calculate the mean monthly rate we captured trends in eligible populations and corresponding coverage, we included monthly patterns in 2017 through to 2022.

## Data analysis

We reviewed pre- and post-COVID-19 immunization coverage trends. To examine trends in both periods (pre and post-COVID-19) and estimate an effect size (change in slope due to COVID-19) we employed segmented quasi-Poisson regression analysis. We included an offset variable to transform the result into a rate; and to account for population fluctuations over time we age-standardized our sample in person-years given the relatively instability over time [29]. To account for seasonal impacts that could create consistent highs and lows in data we used seasonal models with harmonic terms that take seasonal factors into account [24–31]. There are various ways of adjusting for seasonality-here we use harmonic terms specifying the number of sin and cosine pairs to include (in this case 2) and the length of the period (12 months) and this has been reported elsewhere. Owing to the time sequencing of data points used in time series analysis, residual autocorrelation can lead to the violation of regression assumptions. Where significant residual autocorrelation was detected ($P < .10$) and the assumptions of the general linear models become problematic, robust standard errors were generated (using a sandwich estimator) to produce more conservative estimates of uncertainty. R statistical software (version 3.3.2; RStudio, Inc.) was used for all analyses (version 1.0.136; RStudio, Inc). Statistical significance was taken as $P < .05$.

The equation that were used in this ITS analysis:

$Yt = b0 + b1T + b2Xt + b3TXt + b4\sin((2pi/12)*m) + b5\cos((2pi/12)*m) + b6\sin((4pi/12)*m) + b7\cos((4pi/12)*m)$ Where b0 represents the baseline level at $T = 0$, b1 is interpreted as the change in outcome associated with a time unit increase (representing the underlying pre-COVD-19 trend), b2 is the level change following the COVID-19 period, b3 indicates the slope change following the COVID-19 (using the interaction between time and COVID-19: TXt), m is the calendar month, and b4-b7 are coefficients on the seasonal adjustment terms [29].

## Results

Between 2017 and 2020 February, the DPT 2, 3 and OPV 2, 3, adjusted rates per 100,000 population dropped significantly, as evidenced by slope change with all p-values for each vaccine being less than 0.05. Monthly Measles 2 vaccinations slope change pre-COVID-19 was 0.998 (0.995–1.003), the slope change during COVID-19 was 1.00 (0.997–1.010), and with a significant step change of about 41% (RR = 0.59, 95%CI: 0.43–0.80). We did not observe significant shifts in the coverage of, OPV 1, DPT 2 and Measles 1 in pre-pandemic period; however, a huge drop was seen in the coverage measles 2 during the COVID-19 period. Another significant drop was observed in Rotavirus vaccine 1 and 2; and was not surprising owing to the shortage of the vaccine nationally during the period of observation. No difference was

**Table 2. Segmented regression results showing Relative Risk (RR) for COVID-19, pre-intervention and post-intervention.**

| Type of vaccine | Mean Monthly Count | | Mean Monthly Rate per 100, 000 | | Slope Change pre-COVID-19 | PValue[b] | Slope Change during-COVID-19 | PValue[b] | Step Change | P-Value |
|---|---|---|---|---|---|---|---|---|---|---|
| | Before COVID-19 | After COVID-19 | Before COVID-19 | After COVID-19 | RR (95% CI)[C] | | RR(95%CI)[C] | | RR (95% CI)[C] | |
| BCG | 55, 123 | 58, 103 | 101,108 | 100,782 | 0.997(0.994–0.997) | 0.126 | 1.001(1.000–1.011) | 0.016 | 0.78(0.63–0.97) | 0.028 |
| Measles dose 1 | 53, 465 | 56, 202 | 98,041 | 97,498 | 0.998(0.995–1.003) | 0.609 | 1.003(0.998–1.010) | 0.253 | 0.85(0.65–1.01) | 0.226 |
| Measles dose 2 | 53, 465 | 39,449 | 98,041 | 68,435 | 0.998(0.995–1.003) | 0.620 | 1.00(0.997–1.010) | 0.277 | 0.59 (0.43–0.80) | 0.0001 |
| DPT 1 | 5, 088 | 58, 054 | 99,206 | 100,713 | 0.998(0.995–1.000) | 0.096 | 1.006(1.001–1.008) | 0.034 | 0.84(0.72–0.99) | 0.043 |
| DPT 2 | 52, 915 | 56, 623 | 97,057 | 98,232 | 0.997(0.995–0.998) | 0.0001 | 1.005(1.002–1.007) | 0.121 | 0.84(0.72–0.98) | 0.028 |
| DPT 3 | 51,148 | 54, 823 | 93,815 | 95,113 | 0.997(0.995–0.998) | 0.034 | 1.004(1.001–1.008) | 0.007 | 0.85(0.73–0.99) | 0.044 |
| Rota 1 | 53, 633 | 44, 010 | 98,366 | 76,716 | 0.998(0.992–1.005) | 0.578 | 0.97(0.96–0.98) | 0.0001 | 5.29(3.24–8.62) | 0.0001 |
| Rota 2 | 52, 264 | 42, 958 | 95,858 | 74,889 | 0.998(0.992–1.004) | 0.558 | 0.97(0.96–0.98) | 0.0001 | 5.31(3.29–8.59) | 0.0001 |
| Oral poliovirus vaccines (OPV) 1 | 50, 751 | 58, 498 | 93,082 | 101,469 | 0.999(0.994–1.004) | 0.764 | 1.004(0997–1.011) | 0.311 | 0.95(0.66–1.24) | 0.543 |
| Oral poliovirus vaccines (OPV) 2 | 52, 456 | 57, 066 | 96,225 | 98,979 | 0.997(0.995–1.000) | 0.017 | 1.007(1.004–1.011) | 0.0001 | 0.76(0.65–0.90) | 0.001 |
| Oral poliovirus vaccines (OPV) 3 | 50, 215 | 55, 012 | 92,110 | 95,423 | 0.997(0.994–1.000) | 0.027 | 1.007(1.002–1011) | 0.001 | 0.79(0.66–0.95) | 0.011 |
| Fully immunized | 51, 477 | 55, 472 | 94,361 | 96, 292 | 1.001(0.996–1.005) | 0.811 | 1.997(0.991–1.005) | 0.500 | 1.13(0.83–104) | 0.432 |

[a] Abbreviation: RR, relative risk.

[b] Robust SEs are reported following statistically significant Breusch-Godfrey and Seasonal Breusch-testes for autocorrelation.

[c] Effects adjusted for seasonality using harmonic terms.

observed through all the six doses of OPV one, measles one, BCG and Pentavalent during the pandemic period. However, there was a slight increase in vaccinations in OPV 2 and 3 during the COVID-19 pandemic. The increase in this could be due to Zambia receiving notification of the detection of wild polio viruses type1 (WPV1) in Malawi and Mozambique. In view of this epidemic, Zambia responded with series of supplementary polio immunization activities (SIA) to protect the children against polio, and heightened AFP surveillance activities to promote early detection of possible silent circulation of polio virus transmission. And when put together in the variable of "fully immunized", no significant difference was seen for both the crude mean count and the adjusted rates before and after COVID (see Table 2).

When we reviewed the monthly average immunization coverage over the time of this study, we observed clear downward trends in all the antigens. The declines in BCG, OPV and Pentavalent vaccines were suttle, whilst those in Rotavirus vaccine, and Measles were very dramatic. Coverage of rotavirus vaccination already had a steep declining trend even before the COVID-19 pandemic began, and indeed declined drastically towards zero by the end of 2022 (from 91, 181 to 69, 483 for Rota 1 and 88, 851 to 67, 838 for Rota 2 per 100, 000 population. With the advent of COVID-19, coverage for Measles dose 2 was most impacted with the monthly figures dropping from 90, 873 to 63, 797 per 100, 000 population immediately after. Interestingly, a

fairly stable coverage was then maintained through the rest of the COVID-19 period under study in BCG, DPT 1, 2 and OPV 2, 3.

## Discussion

We present evidence of COVID-19 impact on utilization of child administered vaccines in Zambia using the health facility-level monthly reported outpatient data. This is the first effort to review the negative effects of the pandemic using interrupted time series analysis in Zambia.

While we could not find statistically significant reductions in the monthly BCG, OPV and Pentavalent immunization coverages, we visualize a steady decline over the six years study time. Notably, these vaccines which are all administered early in infanthood were generally well utilized in spite of the pandemic. The finding is consistent with other studies that determined that neither during the research period nor before and after the pandemic, was there any convincing evidence of a major change in vaccination rates for BCG [32–34]. This, is likely because of the strong belief and adherence to early infant immunizations when mothers consistently attend child wellness activities [35–37]. It is very likely that the adherence to these early immunization opportunities may be why there has been a great reduction in under five mortalities in the last decade [38–40]. Here-in, we reported on the sharp decline in rotavirus vaccination, coinciding with a global shortage of these vaccines due to COVID-19 pandemic [41,42]. Anecdotally, during the latter part of 2022, nearly all facilities in Zambia were stocked out of this important vaccine Indeed our findings are consistent with reports from elsewhere that countries generally reported major disruptions to more than 50% of health services, including services for routine vaccinations [13–43].

Our current results show reduced monthly Measles dose two vaccination rates following the advent of COVD-19. The monthly average dropped by a third, from 90,873 to 63,797 with a statistically significant step change risk ratio (0.59 (0.43–0.80; P = 0.0001). These results are consistent across the region with reported monthly average number of vaccination doses administered decreased in thirteen out of the fifteen countries in Africa, with six countries experiencing a reduction of more than 10% due to disruption as a result of the COVID-19 pandemic [44,45]. Similarly, there was a steady decline in measles dose one coverage which would be important to observe into 2023. However, the greater impact was seen with measles dose two in which we had an immediate drop in coverage with the advent of COVID-19. While we normally expected measles dose two coverage to be lower than dose one because it is administered much later (after 18 months toddler age), this drop within dose two comparison is most likely and certainly made worse for dose two during the pandemic as a result of fear of COVID-19 among mothers and hence affected health services utilization. In India during the pandemic, it was discovered that women had child vaccine utilization hesitation because of reasons including fear of COVID-19 exposure and lockdown [46]. This could be that parents might perceive that the Measles dose two is less important to protect their child than the Measles dose one, and therefore the risk benefit in comparison to COVID-19 exposure may have meant that they were more likely to delay or skip Measles dose two than Measles dose one. Parents who, having raised multiple children over the years, have grown accustomed to their child's healthcare provider and may now downplay the significance of vaccinations [47]. Indeed elsewhere it was reported that the number of children receiving Measles dose one vaccinations each month decreased by more than two standard deviations at some point in the second quarter of 2020 [44]. In Canada were they compared April 2020 to 2019, there was a considerable decrease in monthly vaccination rate of about 9.9% [48]. Another study suggest that getting child vaccinated was difficult because the waiting time at the clinic was not reasonable for some mothers [49].

Not surprisingly, Zambia reported multiple clustered outbreaks of measles, unfortunately with deaths in some regions during 2022 and early 2023 [50]. Outreach services are a major driver of the vaccine coverages in Zambia. A disruption to these services could likely have had a major impact on the coverage and this has been reported elsewhere [42–52]. The reasons for the disruption include realignment of both financial and human resources to fighting the pandemic, and refocusing of partner priorities to the pandemic and pandemic preparedness. However, this was different with other vaccines, such as BCG, OPV and Pentavalent, that showed improved vaccine coverage during the pandemic as a result of targeted supplementary immunization campaigns instituted in the hopes of lifting the vaccine coverage nationally against polio outbreaks, as Zambia is a land-linked country neighboring Mozambique and Malawi, which have experienced polio outbreaks [53]. Policies around leaving a minimum number of staff at health facilities to continue essential services was key to maintaining some gains in the immunization space.

Generally, the pandemic had broader and far-reaching consequences, not only on the individual vaccine coverage numbers [54]. With the stretch on health workers, services were generally tilted towards emergency care for critical patients and the health workforce was overwhelmed [55,56]. Global travel and commodity logistics were grossly affected during the lock downs which resulted in disruptions in the supply chain [57,58]. Thus, for countries like Zambia which depends on imported commodities, the negative impact of the pandemic is rather obvious. These facts, together with the unaccounted for excess morbidity and mortality [59] justify observational studies such as ours here. While the COVID-19 pandemic received the much needed public attention, it is important to consider any potential health impacts and other adverse effects on child immunizations. Here we provide a nuanced review of the immunization numbers as they were experienced in Zambia before and during the pandemic.

However, the methods we employed have some short comings: We utilized routine data which we know has some gaps within and therefore error-prone especially that collected shortly after the pandemic set in. The potential for concurrently occurring events to skew estimates of intervention effects is one possible drawback of interrupted time series designs. Secondary data were employed in the study. There was therefore no room to add new variables to control for other confounding in the regression analysis such as factors that influence mothers' behaviors towards child immunization. Also, because this study employed aggregated data from national level, it was unable to determine whether there were inequalities based on geography, socioeconomic status and race.

## Conclusion

In conclusion, the COVID-19 pandemic has had a varied consequences on the utilization of children administered vaccines. Although immunization programs were still used, there was a large countrywide decline in immunizations during the pandemic for measles dose two and rotavirus. Proactive and focused measures are required to prevent and mitigate these impacts in upcoming pandemics to ensure the continuity of essential health services. To prevent any negative effects on the uptake of children vaccines, additional cutting-edge measures, such as supplemental vaccination programs, could be implemented in the pandemic response.

## Supporting information

**S1 Fig. Vaccine coverage before and after COVID-19 pandemic.** Data points represent monthly rates of BCG between 2017 and 2022. Gray shaded area depicts the onset of COVID-19 pandemic in Zambia. Dashed lines represent fitted estimates using a linear step change

model. The curved lines represents fitted values for seasonally adjusted models.
(DOCX)

**S2 Fig. Vaccine coverage before and after COVID-19 pandemic.** Data points represent monthly rates of OPV dose 1, 2 & 3 between 2017 and 2022. Gray shaded area depicts the onset of COVID-19 pandemic in Zambia. Dashed lines represent fitted estimates using a linear step change model. The curved lines represent fitted values for seasonally adjusted models.
(DOCX)

**S3 Fig. Vaccine coverage before and after COVID-19 pandemic.** Data points represent monthly rates of Rotavirus dose 1 and 2 between 2017 and 2022. Gray shaded area depicts the onset of COVID-19 pandemic in Zambia. Dashed lines represent fitted estimates using a linear step change model. The curved lines represents fitted values for seasonally adjusted models. There was a drop owing to the shortage of the vaccine nationally during the period of observation.
(DOCX)

**S4 Fig. Vaccine coverage before and after COVID-19 pandemic.** Data points represent monthly rates of Pentavalent dose 1, 2 & 3 between 2017 and 2022. Gray shaded area depicts the onset of COVID-19 pandemic in Zambia. Dashed lines represent fitted estimates using a linear step change model. The curved lines represents fitted values for seasonally adjusted models. Vaccine coverage before and after COVID-19 pandemic.
(DOCX)

**S5 Fig. Vaccine coverage before and after COVID-19 pandemic.** Data points represent monthly rates of Measles dose 1 and 2 between 2017 and 2022. Gray shaded area depicts the onset of COVID-19 pandemic in Zambia. Dashed lines represent fitted estimates using a linear step change model. The curved lines represents fitted values for seasonally adjusted models.
(DOCX)

**S6 Fig. Vaccine coverage before and after COVID-19 pandemic.** Data points represent monthly rates of and Fully Immunized between 2017 and 2022. Gray shaded area depicts the onset of COVID-19 pandemic in Zambia. Dashed lines represent fitted estimates using a linear step change model. The curved lines represents fitted values for seasonally adjusted models.
(DOCX)

**S1 Table. Data Completeness from 2017 to 2018 by month.**
(DOCX)

**S2 Table. Segmented regression results showing Relative Risk (RR) for COVID-19, pre-intervention and post-intervention.**
(DOCX)

**S1 File. R code used for the analysis.**
(R)

## Author Contributions

**Conceptualization:** Kelvin Mwangilwa, Nyambe Sinyange, Roma Chilengi.

**Data curation:** Charles Chileshe, Roma Chilengi.

**Formal analysis:** Kelvin Mwangilwa, Isaac Fwemba.

**Investigation:** Kelvin Mwangilwa.

**Methodology:** Kelvin Mwangilwa, Isaac Fwemba, Roma Chilengi.

**Software:** Charles Chileshe.

**Writing – original draft:** Kelvin Mwangilwa.

**Writing – review & editing:** Kelvin Mwangilwa, Charles Chileshe, John Simwanza, Musole Chipoya, Davie Simwaba, Nathan Kapata, Mazyanga Lucy Mazaba, Nyuma Mbewe, Kapina Muzala, Nyambe Sinyange, Roma Chilengi.

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
