## [Decision Letter · Decision Letter 0]

29 Jun 2023

PGPH-D-23-00790

Evaluating the Impact of COVID-19 on routine childhood immunizations coverage in Zambia

Dear Dr. Mwangilwa,

Thank you for submitting your manuscript to PLOS Global Public Health. After careful consideration, we feel that it has merit but does not fully meet PLOS Global Public Health’s publication criteria as it currently stands. Therefore, we invite you to submit a revised version of the manuscript that addresses the points raised during the review process.

The manuscript has been evaluated by two reviewers, and their comments are available below.

The reviewers have raised a number of concerns that need attention. They request several revisions to improve the reproducibility of the Methods and Analysis, as well as the presentation of the conclusions.

Please note that we expect all researchers with submissions to PLOS in which author-generated code underpins the findings in the manuscript to make all author-generated code available without restrictions upon publication of the work. Please see hhttps://journals.plos.org/globalpublichealth/s/materials-software-and-code-sharing#loc-sharing-code for details.

Could you please revise the manuscript to carefully address the concerns raised?

One or more reviewers has recommended that you cite specific previously published works. As always, we recommend that you please review and evaluate the requested works to determine whether they are relevant and should be cited. It is not a requirement to cite these works. We appreciate your attention to this request.

We look forward to receiving your revised manuscript.

Kind regards,

Marianne Clemence

Staff Editor

Journal Requirements:

1. We noticed that you used "unpublished data" in the manuscript. We do not allow these references, as the PLOS data access policy requires that all data be either published with the manuscript or made available in a publicly accessible database. Please amend the supplementary material to include the referenced data or remove the references.

2. In the online submission form, you indicated that "The corresponding author can be reached for a reasonable request for the datasets utilized in the current study". All PLOS journals now require all data underlying the findings described in their manuscript to be freely available to other researchers, either 1. In a public repository, 2. Within the manuscript itself, or 3. Uploaded as supplementary information.

Additional Editor Comments (if provided):

Reviewers' comments:

Reviewer's Responses to Questions

**Comments to the Author**

1. Does this manuscript meet PLOS Global Public Health’s publication criteria? Is the manuscript technically sound, and do the data support the conclusions? The manuscript must describe methodologically and ethically rigorous research with conclusions that are appropriately drawn based on the data presented.

Reviewer #1: Partly

Reviewer #2: Yes

2. Has the statistical analysis been performed appropriately and rigorously?

Reviewer #1: I don't know

Reviewer #2: Yes

3. Have the authors made all data underlying the findings in their manuscript fully available (please refer to the Data Availability Statement at the start of the manuscript PDF file)?

Reviewer #1: No

Reviewer #2: No

4. Is the manuscript presented in an intelligible fashion and written in standard English?

Reviewer #1: Yes

Reviewer #2: Yes

5. Review Comments to the Author

Reviewer #1: Overview: In this article, the authors analyze trends in routine immunization coverage in Zambia using an interrupted time series approach, comparing pre-pandemic and pandemic-era immunization rates. This topic is of substantial interest as immunization programs analyze their response and resilience during the COVID-19 pandemic and assess challenges that may lie ahead. I have a number of comments below, intended to help better understand the analysis that was conducted and some of the results, as well as some suggestions for opportunities to frame the discussion in light of the authors’ key findings. I thank the authors for undertaking this important analysis and hope that these comments will help to strengthen the manuscript.

Major comments:

1. The authors use DHIS2 data to determine immunization coverage as mentioned on page 3, under “data sources”. It would be helpful for the authors to provide a bit more detail about how these data were used. For instance:

1a. What population estimates / denominators were used to estimate coverage from the number of doses delivered, and did those change from month to month / year to year?

1b. Were there any issues with incomplete reporting from health facilities, and/or did the completeness of reporting change during any part of the pandemic? Some analysis or discussion of the completeness would be helpful, as there were concerns (especially early in the pandemic) that diversion of workforce to COVID-related activities may compromise data quality and completeness.

2. The analysis plan (page 3) could be of interest to other groups or countries who plan to undertake similar analyses. I have a few questions:

2a. To facilitate further evaluation of the methods, could the authors include the equations that were used in this analysis (i.e. for the regression model) including how they constructed the seasonal/harmonic terms?

2b. Will the authors plan to deposit the code in a public repository? It could be quite useful for others to use as a template for their own analyses.

3. The description of the methods on page 3 suggests that the methods estimate a change in slope due to COVID-19, but I’m not sure that I see that happening in the models shown in the various figures. I see that in Table 1 the authors list “trend change”, which I am assuming is the change in the slope. But it seems very odd to me that all the trend changes are essentially 1, even in cases – like rotavirus – where different slopes would seem to allow the model to fit much better than it currently does. If the authors’ current model does theoretically allow the slope to vary, do the authors have any explanation for why all of the slopes are essentially 1?

For instance, Figure 3 shows the model fit to Rota1 and Rota2 coverage. If the model were really fitting a different slope for each time period, I would expect the post-pandemic period to have a much more negative slope than the pre-pandemic period.

Instead, the inclusion of the stockout months in 2022 results in a negative slope that seems to be basically the same for both time periods. In the pre-pandemic months, this slope overestimates the trend in the decline in RotaC, and then underestimates the trend in decline in the post-pandemic period. This also means that the model artificially shows an increase in coverage at the start of the pandemic, when in reality the coverage was about the same, and the modeled “increase” in coverage between March / April 2020 is just an artifact of an overly negative slope in the pre-pandemic time period.

If the authors haven’t tried this already, I would suggest that they allow the slope to vary between the two periods, as I think that this may improve the model fit (Rota being an obvious example, but also potentially for other vaccines. If the model is already trying to do this, then it would be good to understand why all the “trend change” is essentially always 1 (same slope before and during pandemic), and with such little uncertainty around that estimated parameter. Perhaps something has gone wrong in the model specification or the model is not converging?

Last, if the authors think that Rota stockouts were not related to the COVID pandemic, then they might also consider excluding these months from the analysis. If they think that the COVID pandemic played a role in stockouts, e.g. via global supply chain issues – then it might be appropriate to retain these months in the analysis. Regardless, however, the fact that the model is not capturing the clear difference in slopes between the pre- and post-pandemic periods needs further exploration and/or explanation.

4. In the discussion (page 15) the authors write “Indeed our findings are consistent with reports from elsewhere that countries generally reported major disruptions to more than 50% of health services, including services for routine vaccinations [9][30][31].” The authors also write in the conclusion (page 16): “Although immunization programs were still used, there was a large countrywide decline in immunizations during the pandemic.” In general, however, the authors actually find no major effects of the pandemic for most vaccines, aside from MCV2 – or are the authors also considering the Rotavirus stockout to be part of the pandemic? In either case, the authors may want to be more specific about which vaccines were affected and which were not in these sections of the manuscript.

5. In the discussion (page 15), the authors write about the disruptions shown to MCV2 immunization, which are dramatic and persistent. There is some discussion about how this disruption is consistent with other data on first-dose measles disruptions elsewhere. In this analysis, however, the authors show major disruptions to MCV2 but not MCV1. My major question is: why was MCV2 more greatly disrupted than MCV1 – do the authors have any hypotheses that they could share? Is this related to any specific effects of the pandemic on the delivery of vaccines to older children? Were there any changes to policy in Zambia that could have adversely affected the delivery of MCV2 but not MCV1? (The coverage of infant vaccines seems to be otherwise relatively well preserved, aside from rotavirus stockouts).

6. More broadly, the authors find primarily that, for most vaccines, there were no significant changes in coverage due to the pandemic – but there were longer-term declines in coverage that preceded the pandemic and persisted throughout the pandemic. These findings are not explored in much detail in the discussion, and it would be helpful to have some interpretation of these findings.

6a. What do the authors think is driving the longer-term declines in coverage in Zambia, which are certainly troubling?

6b. Do the authors have any insight into the strategies that were used to maintain immunization services during the pandemic in Zambia? The ability of the country to maintain immunization delivery for many vaccines during the pandemic is notable from these results. Knowing how the country achieved this would be very valuable!

Minor comments:

1. Have the authors seen the recently published analysis by Winter et al examining the early impact of the COVID-19 pandemic on immunization coverage in Zambia, also published in PLOS GPH? (https://doi.org/10.1371/journal.pgph.0000554, Winter AK, Takahashi S, Carcelen AC, Hayford K, Mutale W, Mwansa FD, Sinyange N, Ngula D, Moss WJ, Mutembo S. An evaluation of the early impact of the COVID-19 pandemic on Zambia's routine immunization program. PLOS Glob Public Health. 2023 May 2;3(5):e0000554). This may have been published after the authors prepared their manuscript, and I think that the objectives and time scale covered are different enough that the author’s manuscript is still sufficiently novel. (The other analysis looked only at 2020). Still, it may be worth referencing this manuscript and comparing the two studies’ findings for 2020.

2. In the abstract, I wonder if some of the text is missing from the following sentences: “A rise in Rotavirus dose one (RR,1.29; 95% CI: 1.02-1.63, P<. 0.030) with a slop of (RR, 0.98; 95% CI: 0.97-0.99, P<. 0.001), Rotavirus dose two (RR, 1.30; 95% CI: 1.13-1.49, P<. 0.0002) with a slop of (RR, 0.98; 95% CI: 0.98-0.99, P<0.0001) and BCG (RR, 1.00; 95% CI: 0.99- 1.00, P<.0.001).”

2a. Should “slop” be “slope”?

2b. What do the relative risks mean here? Some appear to relate to the intercepts, and some to the slopes. I would suggest being consistent in reporting either slope, intercept, or both. Also, relative risk is a little bit of a challenging term to use here. I might suggest phrasing these more consistently in terms of “intercepts” (which reflect whether or not there appea

---

## [Decision Letter · Decision Letter 1]

15 Jan 2024

PGPH-D-23-00790R1

Evaluating the Impact of COVID-19 on routine childhood immunizations coverage in Zambia

Dear Kelvin Mwangilwa,

Thank you for submitting your manuscript to PLOS Global Public Health. After careful consideration, we feel that it has merit but does not fully meet PLOS Global Public Health’s publication criteria as it currently stands. Therefore, we invite you to submit a revised version of the manuscript that addresses the points raised during the review process.

We look forward to receiving your revised manuscript.

Kind regards,

Nyamongo Onkoba, Ph.D.

Academic Editor

Journal Requirements:

Additional Editor Comments (if provided):

Reviewers' comments:

Reviewer's Responses to Questions

**Comments to the Author**

1. If the authors have adequately addressed your comments raised in a previous round of review and you feel that this manuscript is now acceptable for publication, you may indicate that here to bypass the “Comments to the Author” section, enter your conflict of interest statement in the “Confidential to Editor” section, and submit your "Accept" recommendation.

Reviewer #1: (No Response)

2. Does this manuscript meet PLOS Global Public Health’s publication criteria? Is the manuscript technically sound, and do the data support the conclusions? The manuscript must describe methodologically and ethically rigorous research with conclusions that are appropriately drawn based on the data presented.

Reviewer #1: Yes

3. Has the statistical analysis been performed appropriately and rigorously?

Reviewer #1: Yes

4. Have the authors made all data underlying the findings in their manuscript fully available (please refer to the Data Availability Statement at the start of the manuscript PDF file)?

Reviewer #1: (No Response)

5. Is the manuscript presented in an intelligible fashion and written in standard English?

Reviewer #1: Yes

6. Review Comments to the Author

Reviewer #1: I appreciate the authors’ detailed responses to my previous comments and edits to the manuscript. I have a few smaller follow-up comments below, and I note that unfortunately one of the results tables is cut off in the PDF version that I have access to – it would be helpful to be able to review the last few columns of the table. Overall, however, the manuscript is significantly strengthened by the authors’ revisions, and my thanks to the authors for all their efforts.

Major comments:

1. Previous major comment #1, regarding DHIS2 data. Many thanks to the authors for their reply and for the additions to the manuscript regarding population estimates and data completeness, which are very helpful. The 94% reporting completeness is quite impressive and the addition of Table 1 helps to show that reporting has been sustained at this high level (and even increasing over the last 5 years) even during the pandemic. I have no further comments in this area.

2. Regarding the analysis plan: Thanks to the authors for adding the description of the harmonic terms (that two sin and cosine pairs were added), and for the addition of the equations used in the analysis. This description is very helpful. One additional clarification: did the authors use a two-step process, where they first adjusted for seasonality using the harmonic terms, and then used the seasonally-adjusted T as the input into the ITS model? Or did the ITS model itself contain the harmonic terms? Either approach could be reasonable, but I would suggest that the authors clarify.

My sense from the description is that a two-step process was used, and in that case, it would be sufficient just to add a brief statement explaining that this was a two-step process (and that T in the ITS analysis is seasonally adjusted – perhaps adding a note to this effect under point (i) in the equation description). But if the seasonal terms were included in the ITS regression itself (e.g. if the equation for the ITS regression was truly Yt = b0 + b1T + b2sin(2pi/12*T) + b3cos(2pi/12*T)… ), then it would be useful to re-write the ITS equation to fully show these sin/cosine harmonic terms as well.

Thanks also to the authors for willingness to deposit the code in a repository. No further questions in this area.

3. Previous major comment #3, regarding the interpretation of changes in slope / trends. Thanks to the authors for clarifying in the text. The addition of the ITS equations also is helpful in clarifying how to interpret these results. A few additional comments:

a. Note that the previous Table 1 (Segmented regression results…) should now be re-labeled as Table 2 as a new Table 1 has been added

b. The “Segmented regression results” table is cut off in the PDF version of the file that I have access to, unfortunately – can it be resized so that we can see the additional columns? In particular, I am not sure what the additional RR column that is cut of represents – does this show whether the slopes in the pre- and during-COVID periods are significantly different? Or does it show whether the level is different (i.e. the intercept b2?) Ideally I think that this table would show all of these – i.e. (1) the trends (slopes) both pre-COVID and during COVID, (2) whether the slopes are different before and after COVID, and (3) whether the levels (intercept) are different before and after COVID. Perhaps it already does, but I can’t see it because the table was cut off in the PDF conversion?

c. The RR (95% CI) header for the slope change pre-covid-19 column in the “segmented regression results” table doesn’t have a superscript (c), indicating that the effects are adjusted for seasonality, but the RR column for the slope change during COVID-19 does have that superscript. Should both have the superscript if both are adjusted for seasonality?

4. Previous major comment #4, regarding the discussion and clarifying that measles (2nd dose) and rotavirus were those affected. I appreciate the authors’ clarifications in the discussion, no further comments.

5. Previous major comment #4, regarding the specific disruptions to MCV2 coverage. Thanks to the authors for their additions to the manuscript. The authors propose that these drops may be due to fear of COVID-19 among mothers and decreased health service utilization. The manuscript doesn’t quite specify, however, why this would have affected MCV2 more than MCV1. Could the authors add a sentence explaining why they think that these factors affected MCV2 more than MCV1? (for instance, is this because parents might perceive that the MCV2 dose is less important to protect their child than the MCV1 dose, and therefore the risk/benefit calculations in comparison to COVID exposure may have meant that they were more likely to delay or skip MCV2 than MCV1? Or are there other mechanisms?)

6. Previous major comment #6, regarding a broader discussion of trends in coverage and strategies used in Zambia. The authors’ added paragraph in the discussion is very much appreciated and provides excellent context for their findings; no further comments in this area.

Minor comments:

Previous minor comment #2, regarding the abstract. Thanks to the authors for the changes, which I think are very helpful. I note that the third sentence of the results still seems to have some words that are missing. Should this read “A decreased slope **was observed** in Rotavirus dose one (RR, 0.966; 95% CI: 0.957-0.976) and Rotavirus dose two (RR, 0.966; 95% CI: 0.950-0.983).”?

Previous minor comment #3: Thanks to the authors for their response. I still see “childcare services” in the Introduction, in the sentence “Concerns about an increase in morbidity and mortality for illnesses other than COVID-19, particularly for childcare services, are being raised as a result of these pauses in the use of health services”. Would suggest that this be changed to “child health”.

All other previous minor comments have been addressed. My thanks to the authors.

7. PLOS authors have the option to publish the peer review history of their article (what does this mean?). If published, this will include your full peer review and any attached files.

**Do you want your identity to be public for this peer review?** For information about this choice, including consent withdrawal, please see our Privacy Policy.

Reviewer #1: No

---

## [Decision Letter · Decision Letter 2]

27 Mar 2024

PGPH-D-23-00790R2

Evaluating the Impact of COVID-19 on routine childhood immunizations coverage in Zambia

Dear Dr. Mwangilwa,

Thank you for submitting your manuscript to PLOS Global Public Health. After careful consideration, we feel that it has merit but does not fully meet PLOS Global Public Health’s publication criteria as it currently stands. Therefore, we invite you to submit a revised version of the manuscript that addresses the points raised during the review process.

We look forward to receiving your revised manuscript.

Kind regards,

Miquel Vall-llosera Camps, Ph.D.

Staff Editor

Journal Requirements:

Reviewers' comments:

Reviewer's Responses to Questions

**Comments to the Author**

1. If the authors have adequately addressed your comments raised in a previous round of review and you feel that this manuscript is now acceptable for publication, you may indicate that here to bypass the “Comments to the Author” section, enter your conflict of interest statement in the “Confidential to Editor” section, and submit your "Accept" recommendation.

Reviewer #1: (No Response)

2. Does this manuscript meet PLOS Global Public Health’s publication criteria? Is the manuscript technically sound, and do the data support the conclusions? The manuscript must describe methodologically and ethically rigorous research with conclusions that are appropriately drawn based on the data presented.

Reviewer #1: Yes

3. Has the statistical analysis been performed appropriately and rigorously?

Reviewer #1: No

4. Have the authors made all data underlying the findings in their manuscript fully available (please refer to the Data Availability Statement at the start of the manuscript PDF file)?

Reviewer #1: Yes

5. Is the manuscript presented in an intelligible fashion and written in standard English?

Reviewer #1: Yes

6. Review Comments to the Author

Reviewer #1: Thanks to the authors for their revisions. I have two remaining clarifications, again regarding the harmonic / seasonal terms in the ITS analysis.

Question #1:

Originally, the equation was given as:

"Yt =b0 + b1T + b2Xt + b3TXt, Where b0 represents the baseline level at T = 0, b1 is interpreted as the change in outcome associated with a time unit increase (representing the underlying pre-COVD-19 trend), b2 is the level change following the COVID-19 period and b3 indicates the slope change following the COVID-19 (using the interaction between time and COVID-19: TXt)"

I had initially asked about how exactly the seasonality adjustment was performed - was it a two-step process, whereby the harmonic sin/cos pairs were used to generate a seasonally adjusted value for T and then the regression ("Yt =b0 + b1T + b2Xt + b3TXt") was fit? Or were the harmonic terms included directly in the regression equation in a single step?

The authors write that they used a two-step process, but then write the equation as it were a single large regression that was fit in a single step process (following the example that I had written - sorry if the example equation caused confusion!).

The manuscript now reads:

"In standard ITS analyses, the following segmented regression model is used: Yt = b0 + b1T + b2sin(2pi/12*T) + b3cos(2pi/12*T), Where b0 represents the baseline level at T = 0, b1 is interpreted as the change in outcome associated with a time unit increase (representing the underlying pre-COVD-19 trend), b2 is the level change following the COVID-19 period and b3 indicates the slope change following the COVID-19 (using the interaction between time and COVID-19: TXt) [22]."

This isn't correct, however, as the interpretation of the covariates b1, b2, b3, etc. would all change with the revised equation, and no longer would have the same meaning as in the original equation.

From the code provided (line 190), I now see that there was indeed a one-step adjustment used; that is, that the regression equation directly included the harmonic terms. The code is written as:

model3 <- glm(bcg ~ offset(log(stdpop)) + covid19 + time + harmonic(month,2,12),

family=quasipoisson, data)

where harmonic is (I think) from the tsModel package. From that documentation, harmonic(month,2,12) creates two *pairs* of sin/cos terms - is that right? The source code suggests that this is done by generating sin(x*n*(2pi/period)), where x is your variable of interest (in this case, month), and n = 1, 2, 3 ... N (up to the number N of pairs of sin/cos terms that are added). See https://rdrr.io/cran/tsModel/src/R/tsModelSpec.R#sym-harmonic for the underlying code.

If that's true - and I would ask the authors to check this logic closely - then I think that the equation should look something like:

Yt =b0 + b1T + b2Xt + b3TXt + b4sin((2pi/12)*m) + b5cos((2pi/12)*m) + b6sin((4pi/12)*m) + b7cos((4pi/12)*m)

Where b0 represents the baseline level at T = 0, b1 is interpreted as the change in outcome associated with a time unit increase (representing the underlying pre-COVD-19 trend), b2 is the level change following the COVID-19 period, b3 indicates the slope change following the COVID-19 (using the interaction between time and COVID-19: TXt), m is the calendar month, and b4-b7 are coefficients on the seasonal adjustment terms.

Is that right? Sorry again for the confusion, and please check this logic closely to make sure it matches the actual analysis that was conducted. I think that it is important to make sure that the equation is accurately written so that the analysis can be interpreted correctly.

Question #2:

This does bring up one other question: the authors initially wrote that their equation included an interaction term between COVID and time:

"Yt =b0 + b1T + b2Xt + b3TXt, Where b0 represents the baseline level at T = 0, b1 is interpreted as the change in outcome associated with a time unit increase (representing the underlying pre-COVD-19 trend), b2 is the level change following the COVID-19 period and b3 indicates the slope change following the COVID-19 (using the interaction between time and COVID-19: TXt)"

But, as above, the code in line 190 doesn't include this interaction term:

model3 <- glm(bcg ~ offset(log(stdpop)) + covid19 + time + harmonic(month,2,12),

family=quasipoisson, data)

Am I looking at the wrong line in the code, or is there another reason for the discrepancy? Based on the equation in the text, I would have expected to see:

model3 <- glm(bcg ~ offset(log(stdpop)) + covid19 + time + time*covid + harmonic(month,2,12),

family=quasipoisson, data)

I have provided a simplified version of the authors’ code, which just runs model 3 for Rota dose 1 (without the interaction term). This produces results that look quite similar to the authors’ figures (see attached model3.png, which is quite similar to the authors’ Figure 3).

However, when I run a new model (model 4 in the attached code), but include the interaction terms, I see a very different result (see model4.png). As expected, this allows the slope to vary between the COVID and post-COVID periods, with very different interpretations of the ITS analysis.

Can the authors clarify – was the interaction term inadvertently excluded from the analysis here? If so, it seems like the interaction term would need to be included in order to be able to interpret any of the “trend change” results in the manuscript.

Apologies that it took me several revisions to raise this query, as it required me to dive into the code to understand the seasonality terms before I noticed the discrepancy.

7. PLOS authors have the option to publish the peer review history of their article (what does this mean?). If published, this will include your full peer review and any attached files.

**Do you want your identity to be public for this peer review?** For information about this choice, including consent withdrawal, please see our Privacy Policy.

Reviewer #1: No

---

## [Decision Letter · Decision Letter 3]

16 May 2024

PGPH-D-23-00790R3

Evaluating the Impact of COVID-19 on routine childhood immunizations coverage in Zambia

Dear Dr. Mwangilwa,

Thank you for submitting your manuscript to PLOS Global Public Health. After careful consideration, we feel that it has merit but does not fully meet PLOS Global Public Health’s publication criteria as it currently stands. Therefore, we invite you to submit a revised version of the manuscript that addresses the points raised during the review process.

We look forward to receiving your revised manuscript.

Kind regards,

Nnodimele Onuigbo Atulomah, PhD

Academic Editor

Journal Requirements:

2. Please provide separate figure files in .tif or .eps format only and remove any figures embedded in your manuscript file. Please also ensure all files are under our size limit of 10MB.

3. We noticed that you used "unpublished data" in the manuscript. We do not allow these references, as the PLOS data access policy requires that all data be either published with the manuscript or made available in a publicly accessible database. Please amend the supplementary material to include the referenced data or remove the references.

4. We have noticed that you have uploaded Supporting Information files, but you have not included a list of legends. Please add a full list of legends for your Supporting Information files after the references list.

Additional Editor Comments (if provided):

The many rounds of reviews has tremendously improved the manuscript at this level, however the reviewers believe certain minor issues of technical nature and some additional proofreading needs to be performed. I am therefore recommending that the authors carefully review all of the results in the manuscript to make sure that they have been updated for the new analysis. In addition, that the authors should clarify if the interrupted time series is the analysis performed or the study design, as pointed by on of the reviewers. What is in the method section is different from the abstract.

Reviewers' comments:

Reviewer's Responses to Questions

**Comments to the Author**

1. If the authors have adequately addressed your comments raised in a previous round of review and you feel that this manuscript is now acceptable for publication, you may indicate that here to bypass the “Comments to the Author” section, enter your conflict of interest statement in the “Confidential to Editor” section, and submit your "Accept" recommendation.

Reviewer #1: (No Response)

Reviewer #3: All comments have been addressed

2. Does this manuscript meet PLOS Global Public Health’s publication criteria? Is the manuscript technically sound, and do the data support the conclusions? The manuscript must describe methodologically and ethically rigorous research with conclusions that are appropriately drawn based on the data presented.

Reviewer #1: Partly

Reviewer #3: Yes

3. Has the statistical analysis been performed appropriately and rigorously?

Reviewer #1: Yes

Reviewer #3: Yes

4. Have the authors made all data underlying the findings in their manuscript fully available (please refer to the Data Availability Statement at the start of the manuscript PDF file)?

Reviewer #1: Yes

Reviewer #3: Yes

5. Is the manuscript presented in an intelligible fashion and written in standard English?

Reviewer #1: Yes

Reviewer #3: Yes

6. Review Comments to the Author

Reviewer #1: Thanks to the authors for their revisions. The methods are now much easier to follow. I have a very few remaining comments, mainly around ensuring that all of the results of the manuscript have been updated to reflect the new analysis.

1. Did the authors revise all of the results in the manuscript with the new regression results? For instance, the first paragraph of the results states that “Monthly Measles 2 vaccinations was 0.67 times lower during the COVID-19 pandemic (95%CI=0.61-0.74) and declined by 0.998 per month (95% CI=0.997-1.005). However, period between March 2020 and December 2022, Measles 2 vaccines reduced significantly by about 0.2% (RR=0.998, 95% CI: 0.997-1.005).” This is the same as in the previous submission and I am not sure how it relates to the new Table 2, which states that the slope change pre-COVID-19 was 0.998 (0.995-1.003), the slope change during COVID-19 was 1.00 (0.997-1.010), and the step change was 0.59 (0.43 – 0.80). I wonder if the authors have not updated this from the previous analysis with the old, incorrect regression? The discussion of MCV2 in the discussion section appears to have the updated results included, so this may have been just an omission. (Also, the second sentence seems to repeat results in the first sentence, and this could probably be consolidated).

Also, in Table 2, most of the rows appear to have been updated with the new regression results. I believe, however, that the last row (Fully Immunized) has not changed between the previous version of the manuscript and this one. Was this to be expected (e.g. did the fully immunized analysis run for the previous manuscript version already include the appropriate interaction terms)? I mention this just to make sure that it is not inadvertent, e.g. that this row wasn’t updated like the others as the authors made their revisions.

In general, I would suggest that the authors carefully review all of the results in the manuscript to make sure that they have been updated for the new analysis.

2. The manuscript would benefit from some additional proofreading; defer to the editors on this. For instance, there are two mentions of “slop” which should be “slope” in the results section of the abstract.

Reviewer #3: (No Response)

7. PLOS authors have the option to publish the peer review history of their article (what does this mean?). If published, this will include your full peer review and any attached files.

**Do you want your identity to be public for this peer review?** For information about this choice, including consent withdrawal, please see our Privacy Policy.

Reviewer #1: No

Reviewer #3: **Yes: **Titilayo Olaoye

---

## [Editor Report · Decision Letter 4]

5 Jun 2024

Evaluating the Impact of COVID-19 on routine childhood immunizations coverage in Zambia

PGPH-D-23-00790R4

Dear Mr. Mwangilwa,

We are pleased to inform you that your manuscript 'Evaluating the Impact of COVID-19 on routine childhood immunizations coverage in Zambia' has been provisionally accepted for publication in PLOS Global Public Health.

Best regards,

Nnodimele Onuigbo Atulomah, PhD

Academic Editor

All revisions suggested and recommended by the reviewers have been made by the authors to the satisfaction of the editor assigned for editorial process of this submission.